# Enhancing Mitochondrial Function Through Pharmacological Modification: A Novel Approach to Mitochondrial Transplantation in a Sepsis Model

**DOI:** 10.3390/biomedicines13040934

**Published:** 2025-04-10

**Authors:** Bomi Kim, Yun-Seok Kim, Kyuseok Kim

**Affiliations:** Department of Emergency Medicine, CHA University School of Medicine, Seongnam 13488, Republic of Korea; alfks4050@naver.com (B.K.); loupys@naver.com (Y.-S.K.)

**Keywords:** formoterol, mitochondrial transplantation, proinflammation, sepsis

## Abstract

**Background/Objectives:** Sepsis continues to be a significant global health issue, with current treatments primarily focused on antibiotics, fluid resuscitation, vasopressors, or steroids. Recent studies have started to explore mitochondrial transplantation as a potential treatment for sepsis. This study aims to evaluate the effects of enhanced mitochondrial transplantation on sepsis. **Methods:** We examined various mitochondrial-targeting drugs (formoterol, metformin, CoQ10, pioglitazone, fenofibrate, and elamipretide) to improve mitochondrial function prior to transplantation. Mitochondrial function was assessed by measuring the oxygen consumption rate (OCR) and analyzing the expression of genes related to mitochondrial biogenesis. Additionally, the effects of enhanced mitochondrial transplantation on inflammation were investigated using an in vitro sepsis model with THP-1 cells. **Results:** Formoterol significantly increased mitochondrial biogenesis, as evidenced by enhanced oxygen consumption rates and the upregulation of mitochondrial-associated genes, including those related to biogenesis (*PGC-1α*: 1.56-fold, *p* < 0.01) and electron transport (*mt-Nd6*: 1.13-fold, *p* = 0.16; *mt-Cytb*: 1.57-fold, *p* < 0.001; and *mt-Co2*: 1.44-fold, *p* < 0.05). Furthermore, formoterol-enhanced mitochondrial transplantation demonstrated a substantial reduction in TNF-α levels in LPS-induced hyperinflammatory THP-1 cells (untreated: 915.91 ± 12.03 vs. formoterol-treated: 529.29 ± 78.23 pg/mL, *p* < 0.05), suggesting its potential to modulate immune responses. **Conclusions:** Mitochondrial transplantation using drug-enhancing mitochondrial function might be a promising strategy in sepsis.

## 1. Introduction

Sepsis is a life-threatening condition caused by a dysregulated inflammatory response, resulting in organ dysfunction and tissue injury [1,2]. Recent advances in clinical and experimental sepsis research have led to significant discoveries regarding immunometabolism and bioenergetics [3]. Despite the implementation of conventional management approaches such as prompt antibiotic administration and fluid resuscitation, therapeutic interventions remain limited [4]. Thus, identifying novel therapeutic targets in the management of sepsis is crucial.

Mitochondria are multifunctional organelles that play a crucial role in cellular development and energy production [5]. These roles include regulating the biosynthesis of molecules required for cell growth, as well as controlling apoptosis, intracellular calcium levels, redox states, and the immune response [6,7,8,9]. In addition, studies conducted thus far underscore the essential significance of mitochondria in the development of sepsis [10,11]. Increased mitochondrial damage during sepsis has been reported in both laboratory and clinical studies [12,13,14]. Moreover, assessing mitochondrial metabolism through cellular respiration can help identify sepsis patients at a heightened risk of worsening organ failure and effectively monitor their response to treatment [15].

Several studies have explored mitochondrial dysfunction as a potential therapeutic target, and drugs aimed at mitochondrial function have been developed. Formoterol, the United States Food and Drug Administration-approved and highly specific β2-adrenoreceptor agonist, has been shown to stimulate mitochondrial biogenesis in pathological conditions [16]. Elamipretide, also known as SS-31 (D-Arg-Dmt-Lys-Phe-NH2), is a mitochondria-targeting antioxidant. It restores mitochondrial transport and synaptic viability while decreasing the percentage of defective mitochondria, indicating its protective effects against Aβ toxicity [17]. Elamipretide also eliminates reactive oxygen species and increases adenosine triphosphate levels in mitochondria, thereby maintaining mitochondrial membrane potential [18]. Fenofibrate induces mitochondrial reprogramming via carnitine palmitoyltransferase I and the fatty acid oxidation pathway, activates the AMP-activated protein kinase pathway, and inhibits the hexokinase 2 pathway [19]. Metformin, one of the oldest and most widely used antidiabetic drugs, has been shown to promote electron transport chain (ETC) expression in patients with type 2 diabetes and enhance mitochondrial biogenesis [20]. Additionally, pioglitazone is a peroxisome proliferator-activated receptor gamma agonist that exerts beneficial effects on mitochondrial function. Previous studies have demonstrated that pioglitazone enhances mitochondrial structure and bioenergetics in Down syndrome cells, suggesting its potential for improving mitochondrial health [21]. Coenzyme Q10 (CoQ10), a powerful antioxidant, eliminates free radicals and inhibits the initiation and propagation of lipid peroxidation in cellular biomembranes [22]. It also acts as a diffusible electron carrier in the mitochondrial respiratory chain [23].

Previous studies have reported the effects of mitochondrial transplantation in both physiological and pathological states, emphasizing its role in regulating immune responses, maintaining metabolic homeostasis, and supporting other cellular processes [24]. Furthermore, we previously demonstrated that mitochondrial transplantation effectively reversed the negative effects of sepsis in in vivo, ex vivo, and in vitro models [25,26,27].

Figure 1 provides a schematic overview of the experimental approach used in this study. First, we screened drugs known to enhance mitochondrial function and identified formoterol as the most effective in improving mitochondrial health. Additionally, we observed the modulation of hyperinflammatory responses through the transplantation of enhanced mitochondria by formoterol. These findings underscore the therapeutic potential of formoterol-enhanced mitochondrial transplantation in the treatment of sepsis.

## 2. Materials and Methods

### 2.1. Chemicals

Formoterol (F9552), metformin (PHR1084), CoQ10 (C9538), and lipopolysaccharides (LPS, L2762) were purchased from Sigma-Aldrich (St. Louis, MO, USA). Pioglitazone (71745) and fenofibrate (10005368) were purchased from Cayman Chemical (Ann Arbor, MI, USA). Elamipretide (S9803) was obtained from Selleck Chemicals (Houston, TX, USA).

### 2.2. Cell Culture

The rat myoblast L6 cell line was purchased from ATCC (CRL-1458, Manassas, VA, USA). The culture medium consisted of Dulbecco’s Modified Eagle Medium (L0103, DMEM, Biowest, Nuaillé, France) supplemented with 10% heat-inactivated fetal bovine serum (S1480, FBS, Biowest) and 1% penicillin/streptomycin (Gibco, Waltham, MA, USA). Cells at passages 6–8 were used for subsequent experiments. The human monocyte THP-1 cell line was purchased from ATCC (TIB-202). The THP-1 cells were maintained in cell culture flasks using RPMI 1640 medium (L0498, Biowest) supplemented with 10% heat-inactivated FBS and 1% penicillin/streptomycin. All cells were maintained in a 5% CO_2_ atmosphere at 37 °C.

### 2.3. Mitochondrial Isolation and Transplantation

Mitochondrial isolation was performed as previously described [25]. Briefly, L6 cells were homogenized in ice-cold SHE buffer [0.25 M sucrose, 20 mM HEPES, 2 mM EGTA, 10 mM KCl, 1.5 mM MgCl_2_, 0.1% defatted bovine serum albumin (BSA), a protease inhibitor, pH 7.4]. The mitochondria were quantified by measuring protein concentrations using a Bicinchoninic Acid Protein Assay Kit (23227, Thermo Scientific, Waltham, MA, USA). Equal amounts of isolated mitochondria (10 µg per well) from L6 cells were plated into recipient THP-1 cells in the complete medium.

### 2.4. Mitochondrial Respiration Measurement

The Seahorse XF Pro Analyzer (Agilent, Santa Clara, CA, USA) and the XF Cell Mito Stress Test kit (Agilent) were used to measure the oxygen consumption rate (OCR) according to the manufacturer’s protocol. L6 cells were seeded into an XF assay plate at a density of 1 × 10^4^ cells per well and then incubated for 24 h in a 5% CO_2_ atmosphere at 37 °C. Supernatants were replaced with fresh medium containing one of the drugs, and the cells were incubated for an additional 24 h. On the day of the assay, the assay medium was prepared by supplementing XF DMEM with 1 mM pyruvate, 2 mM glutamine, and 10 mM glucose. The cells were washed with assay medium and allowed to equilibrate for 30 min at 37 °C in a non-CO_2_ incubator. Respiratory reagents were loaded into the ports of a sensor cartridge as follows: Port A: oligomycin (1.5 µM, final concentration); Port B: carbonyl cyanide-p-trifluoromethoxyphenylhydrazone (FCCP, 2 µM, final concentration); and Port C: rotenone/antimycin A (0.5 µM, final concentration).

### 2.5. Enzyme-Linked Immunosorbent Assay (ELISA)

L6 cells were seeded at a density of 2 × 10^6^ cells per dish and incubated for 48 h. Subsequently, they were incubated with or without 30 nM formoterol for 24 h. Mitochondria were isolated from cultured cells (as described previously), and they were then added to the THP-1 cells along with the 50 ng/mL LPS. After 24 h of incubation, the supernatants were collected by centrifugation at 2000× *g* for 10 min to remove debris. TNF-α levels were measured using a TNF alpha ELISA Kit (ab181421, Abcam, Waltham, MA, USA) according to the manufacturer’s instructions.

### 2.6. RNA Extraction and Reverse Transcription–Quantitative Polymerase Chain Reaction (RT–qPCR)

L6 cells were treated with 30 nM formoterol for 24 h, after which total RNA was extracted using the AccuPrep^®^ Universal RNA Extraction Kit (K-3140, Bioneer, Daejeon, Republic of Korea). cDNA was synthesized from 1 μg of total RNA using RT PreMix (K-2241, Bioneer), and qPCR was performed using a Real-Time PCR System (788BR10054, BIO-RAD, Hercules, CA, USA). The actin gene was used as the housekeeping control. The relative gene expression was quantified using the 2^−ΔΔCT^ method. Primer sequences used in this experiment are provided in Table 1.

### 2.7. Statistical Analysis

Differences between groups were analyzed using the Kolmogorov–Smirnov normalization test. If both groups tested passed the test, an unpaired two-tailed Student’s *t*-test was used for comparison. Otherwise, the nonparametric Mann–Whitney test was applied. *p*-values < 0.05 were considered statistically significant. Statistical analyses were performed using EZR 4.3.1 (Saitama Medical Center, Jichi Medical University, Saitama, Japan).

## 3. Results

### 3.1. The Drug Treatment Affected Mitochondrial Respiration

Figure 2 shows that mitochondrial respiration was influenced by six drugs: formoterol, metformin, CoQ10, pioglitazone, fenofibrate, and elamipretide. Notably, formoterol increased the basal respiration rate (123.89 ± 23.13 vs. 168.57 ± 23.46 pmol/min) (Figure 2a,b). The maximal respiration rate was also higher in the formoterol-treated group compared to the untreated group (123.89 ± 23.13 vs. 168.57 ± 23.46 pmol/min). In contrast, the other drug-tested groups did not show a comparable positive effect (Figure 2c–g) when using various concentrations of each drug. These results suggest that treatment with formoterol improved mitochondrial function under specific experimental conditions.

### 3.2. Formoterol Increases the Expression of Mitochondrial Genes

To examine whether formoterol modulates the expression of mitochondrial-associated genes, we investigated mRNA levels using RT-qPCR. The presence of formoterol upregulated the expression levels of *PGC-1α* (1.56-fold, *p* < 0.01), *mt-Nd6* (1.13-fold, *p* = 0.16), *mt-Co2* (1.44-fold, *p* < 0.05), and *mt-Cytb* (1.57-fold, *p* < 0.001) (Figure 3). These results further confirmed that formoterol enhances mitochondrial biogenesis.

### 3.3. Formoterol-Enhanced Mitochondrial Transplantation Alleviated the Immune Response

To explore the modulation of the immune system with formoterol-enhanced mitochondria, we examined the levels of TNF-α expression in an in vitro sepsis model induced by LPS. The control group consisted of THP-1 cells that were left untreated. In the 30 nM and 45 nM formoterol^+^/mitochondria^+^ groups, the TNF-α levels were significantly lower compared to the formoterol^-^/mitochondria^-^ group (529.29 ± 78.23, 485.61 ± 36.65 pg/mL vs. 1069.44 ± 14.72 pg/mL, *p* < 0.05, respectively) (Figure 4). Additionally, compared to the formoterol^-^/mitochondria^+^ group, the TNF-α levels in the formoterol^+^/mitochondria^+^ groups were significantly decreased (915.91 ± 12.03 pg/mL, *p* < 0.05, respectively). These findings indicate that formoterol-enhanced mitochondria mitigate LPS-induced proinflammatory responses in THP-1 cells.

## 4. Discussion

We highlight the therapeutic possibilities of improved mitochondrial transplantation in mitigating immune responses, especially within the context of inflammatory conditions like sepsis. Previous research has demonstrated the protective benefits of mitochondrial transplantation in sepsis [28,29]. However, the synergistic effects between the enhanced function of mitochondria and transplantation remain unclear. Initially, we evaluated the effect of formoterol on mitochondrial function. The results of our study show that formoterol enhances OCR in L6 cells and increases the expression of mitochondrial-associated genes such as *PGC-1α*, *mt-Nd6*, *mt-Cytb*, and *mt-Co2*, which supports the role of formoterol in enhancing mitochondrial function. Finally, the transplantation of these enhanced mitochondria led to a reduction in the immune response.

Formoterol has been shown to be a potent inducer of mitochondrial biogenesis [16,30,31] in a range of disease models. For instance, formoterol significantly increased FCCP-uncoupled OCR and mitochondrial DNA copy numbers in primary renal proximal tubule cells and adult feline cardiomyocytes [16]. Furthermore, it affects mitochondrial biogenesis in a renal ischemia–reperfusion injury through the restoration of renal function, protection of renal tubules from injury, and recovery of downregulated mitochondrial protein expression and function [31].

The accurate measurement of cellular OCR is essential for understanding mitochondrial contributions to cellular energy metabolism and bioenergetic function in pathological conditions [32]. The observed increase in OCR following formoterol treatment indicates enhanced mitochondrial respiratory efficiency, consistent with its role in promoting mitochondrial biogenesis and function. Furthermore, OCR quantification serves as a biomarker for assessing mitochondrial-targeted therapies, providing an objective metric for evaluating therapeutic strategies in sepsis [33].

PGC-1α, encoded by the *Ppargc1a* gene, is a critical regulator of mitochondrial biogenesis and a key mediator of mitochondrial energy metabolism [34]. MT-ND6 is a gene that encodes a subunit of complex I in the ETC. This subunit is involved in electron transfer from NADH to ubiquinone. MT-CYB encodes cytochrome b, a core component of complex III. MT-CO_2_ is one of the genes encoding a subunit of cytochrome c oxidase, which catalyzes the reduction of oxygen to water [35,36]. The observed upregulation of these genes following formoterol treatment aligns with enhanced mitochondrial function, as previously reported [37], further supporting the role of formoterol in promoting mitochondrial biogenesis and energy metabolism.

Mitochondrial metabolic status is crucial for regulating inflammatory responses [38]. Mitochondrial transplantation has shown therapeutic potential in an animal model of Parkinson’s disease by providing neuroprotection and reducing neuroinflammation [39]. Moreover, mitochondrial transplantation has been reported to mitigate hyperinflammation and immunosuppression in an in vitro sepsis model [26]. Our findings align with these studies, demonstrating that pharmacologically enhanced mitochondria exhibit improved anti-inflammatory effects, highlighting their potential for therapeutic applications.

Overall, our findings indicate that pharmacological preconditioning of mitochondria might offer superior anti-inflammatory effects compared to conventional mitochondrial transplantation approaches, potentially expanding the therapeutic scope of this strategy. However, several limitations must be acknowledged. To enhance the clinical translation, further research is required to assess the in vivo efficacy of formoterol and its long-term effects on mitochondrial function and inflammatory pathways. Additionally, studies evaluating its impact on other cell types and tissues are essential to validate these results. Lastly, we chose OCR data on optimal drug use, but there might be a chance of mitigating immune dysfunction regardless of OCR data. However, we think that mitochondrial transplantation has an impact on inflammation by enhancing mitochondrial function, so we adopted OCR for the screening method.

## 5. Conclusions

Mitochondrial transplantation using formoterol-enhancing mitochondrial function might be a promising strategy for sepsis.

## Figures and Tables

**Figure 1 biomedicines-13-00934-f001:**
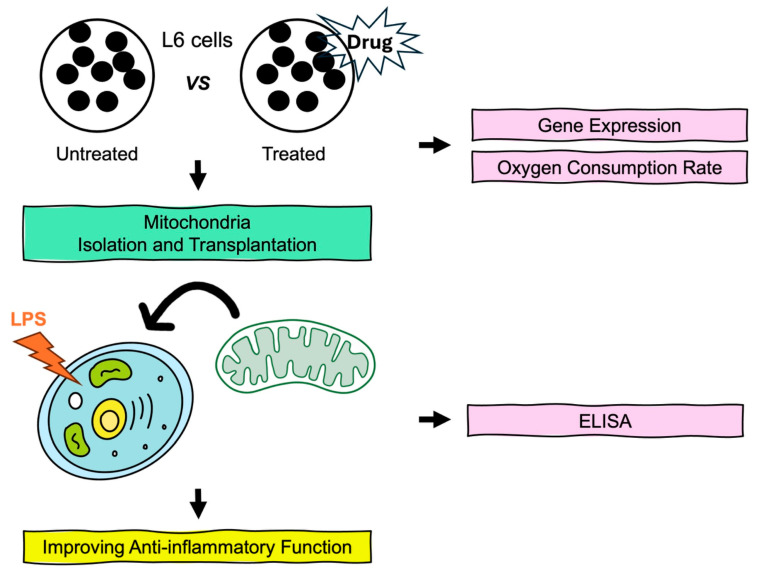
A schematic of the experimental design.

**Figure 2 biomedicines-13-00934-f002:**
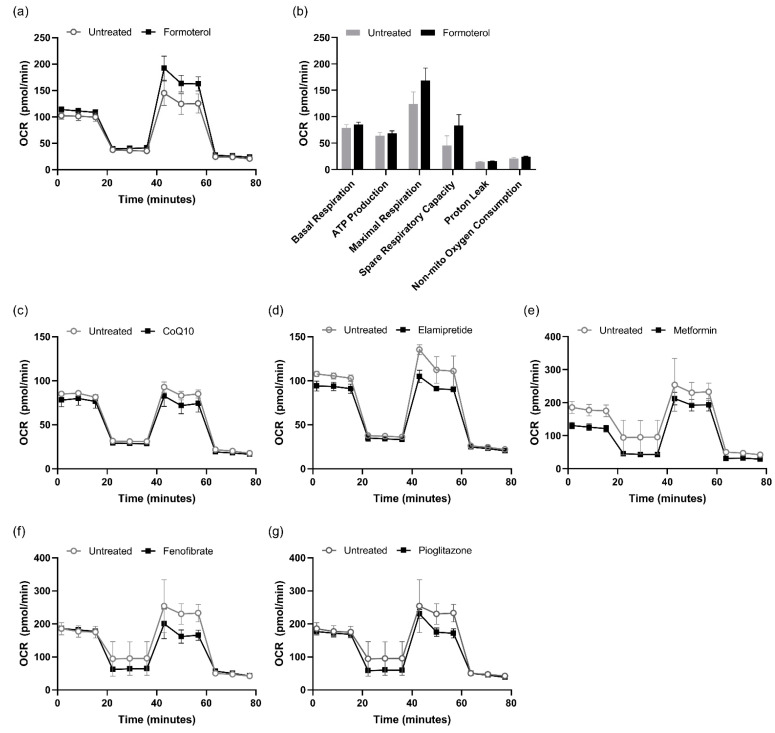
Screening for candidate drugs to enhance mitochondrial function. The OCR was measured in L6 cells following treatment with six different drugs: (**a**) 30 nM formoterol; (**b**) bar graphs showing basal respiration, ATP production, maximal respiration, spare respiratory capacity, proton leak, and non-mitochondrial oxygen consumption; (**c**) 10 nM CoQ10; (**d**) 500 nM elamipretide; (**e**) 1 mM metformin; (**f**) 100 µM fenofibrate; and (**g**) 1 µM pioglitazone. The data are presented as the mean ± SD.

**Figure 3 biomedicines-13-00934-f003:**
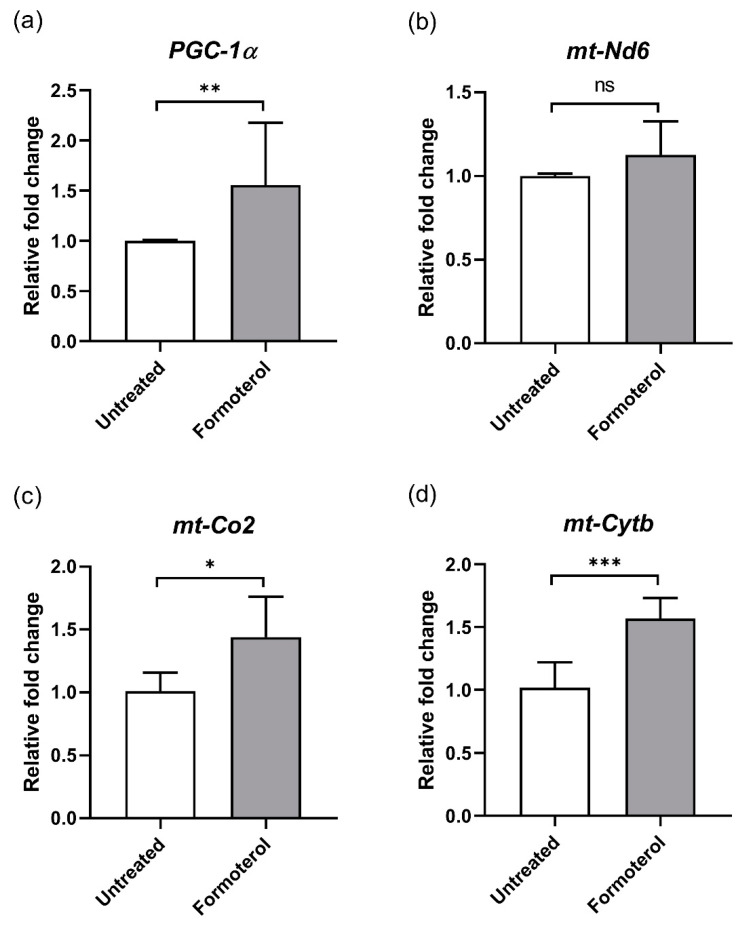
Mechanism of action (MOA) of formoterol on mitochondrial function. Expression analysis of mitochondria-associated genes by RT-qPCR. Relative fold-change in mRNA expression was analyzed for (**a**) *PGC-1α*, (**b**) *mt-Nd6*, (**c**) *mt-Co2*, and (**d**) *mt-Cytb*. *Actb* was used as the reference gene for normalization. Data are presented as the mean ± SD. *: *p* < 0.05, **: *p* < 0.01. ***: *p* < 0.001. ns, not significant.

**Figure 4 biomedicines-13-00934-f004:**
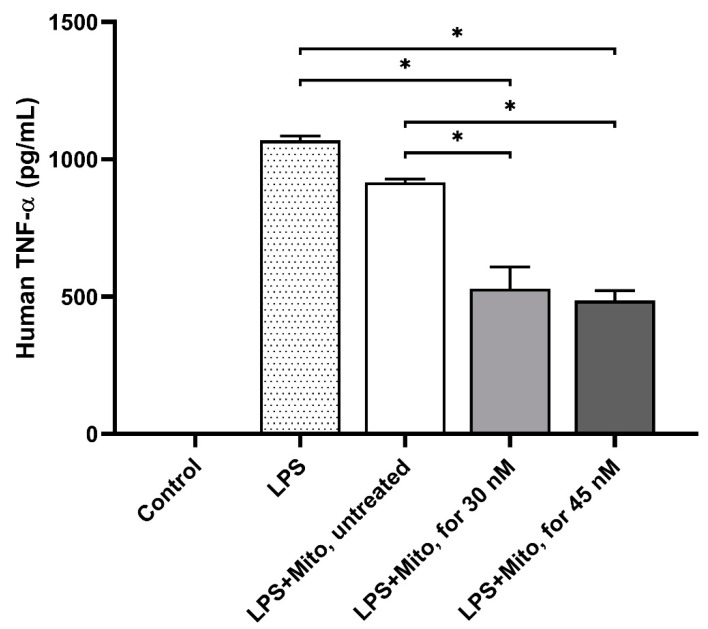
Effects of enhanced mitochondrial transplantation on sepsis. Formoterol-enhanced L6-mitochondrial transplantation showed substantial anti-inflammatory effects in LPS-stimulated THP-1 cells. Data are presented as the mean ± SD. *: *p* < 0.05. Control, THP-1 cells without LPS.

**Table 1 biomedicines-13-00934-t001:** Sequences of primers used for qPCR.

Symbol		Primer Sequences (5’→3’)
*Actb*	Forward	TGTGGATTGGTGGCTCTATC
	Reverse	AGAAAGGGTGTAAAACGCAG
*PGC-1*α (*Ppargc1a*)	Forward	AGGAAATCCGAGCTGAGCTGAACA
	Reverse	GCAAGAAGGCGACACATCGAACAA
*mt-Cytb*	Forward	CCCACAGGATTAAACTCCGA
	Reverse	GTTGGGAATGGAGCGTAGAA
*mt-Co2*	Forward	CAAGACGCTACATCACCTATC
	Reverse	CTAATAGACGAAGTTCACCTGG
*mt-Nd6*	Forward	TAGACCCTCAAGTCTCCGGG
	Reverse	TGGTGGGCTTGGATTGATTGT

## Data Availability

The data presented in this study can be made available upon request from the corresponding author. It will not be open to the public due to further research.

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
