# Peer review of "Enhancing Mitochondrial Function Through Pharmacological Modification: A Novel Approach to Mitochondrial Transplantation in a Sepsis Model"

_biomedicines, 2025, doi:10.3390/biomedicines13040934_

Round 1
Reviewer 1 Report
Comments and Suggestions for Authors
The results of this study might be interesting if the authors will provide evidence that the isolated mitochondria were indeed transplanted into the THP-1 cells.
Crucial information regarding the duration of incubation of the isolated mitochondria with the various drugs, and more importantly - the duration of the incubation of the formoterol-supplemented mitochondria with the LPS-induced hyperinflammatory THP-1 cells prior the assay of TNF-α levels.
Comments on the Quality of English LanguageThere are multiple grammar inaccuracies. The manuscript has to be gone over by an English-proficient person.
Author Response
Reviewer 1
Comments and Suggestions for Authors
The results of this study might be interesting if the authors will provide evidence that the isolated mitochondria were indeed transplanted into the THP-1 cells.
-->We have investigated the effects of mitochondrial transplantation on sepsis model, and we have much evidence to support the mitochondria are transplanted in various conditions including in-vivo and in-vitro model. Other researchers have also shown that the mitochondria have been transplanted in many cell types. In the beginning of the mitochondrial transplantation studies, the direct evidence of mitochondrial transplantation has been provided, but in these days, it is believed in the research society that mitochondrial transplantation do occur, so the direct evidence of transplantation is usually skipped in the paper.
The followings are evidence of our previous studies.
- We used mitochondrial transplantation in in-vivo and in-vitro sepsis model, and showed that mitochondria are transplanted in spleen using Mito Tracker tracking method of the following paper published in Critical Care(2021); The immune modulatory effects of mitochondrial transplantation on cecal slurry model in rat. Additional file 3
- Figure 1 B of the following paper showed the transplanted mitochondria in in-vitro model using Mito Tracker; Delivery of exogenous mitochondria via centrifugation enhances cellular metabolic function, published in Scientific Reports.
- We also showed that mitochondrial genes were effectively transplanted with mitochondrial transplantation of the following paper; Effects of Mitochondrial Transplantation on Transcriptomics in a Polymicrobial Sepsis Model published in IJMS. In that transcriptomics study, the percentage of mitochondrial mapped reads was significantly increased by 1.5-fold (Figure 2b), indicating that our mitochondrial transplantation was technically successful.
- We also have direct evidence that the mitochondria were transplanted into PBMC(not published). In this data using Mito Tracker, mitochondrial transplantation occur more effectively with centrifugation, thereafter, we have used centrifugation for effective mitochondrial transplantation.
Crucial information regarding the duration of incubation of the isolated mitochondria with the various drugs, and more importantly - the duration of the incubation of the formoterol-supplemented mitochondria with the LPS-induced hyperinflammatory THP-1 cells prior the assay of TNF-α levels.
--> We appreciate your important comment. We revised the Methods section to clarify the incubation times as follows:
Line 116-119. L6 cells were seeded into an XF assay plate at a density of 1 x 104 cells per well and then incubated for 24h in a 5% CO2 atmosphere at 37 ℃. The supernatants were replaced with fresh medium containing one of the drugs, and the cells were incubated for an additional 24h.
Line 129-132. Subsequently, they were incubated with or without 30 nM formoterol for 24h. Mitochondria were isolated from cultured cells (as described previously) and they were then added to THP-1 cells along with the 50 ng/mL LPS. After 24h of incubation, the supernatants were collected by centrifugation at 2000 x g for 10 min to remove debris.
Comments on the Quality of English Language
There are multiple grammar inaccuracies. The manuscript has to be gone over by an English-proficient person.
--> We appreciate your good comment. We have reviewed and revised the whole manuscript.

Reviewer 2 Report
Comments and Suggestions for Authors
In the brief report titled “Enhancing mitochondrial function through pharmacological modification: A novel approach to mitochondrial transplantation in a sepsis model” authors describe a strategy to improve the mitochondrial function before transplantation into a cellular model of sepsis. For this strategy, several molecules were tested on rat myoblast L6 cell line and the effect on the oxygen consumption rate (OCR) was evaluated. The best molecule, Formoterol, was used to measure the expression level of mitochondrial-encoded genes in order to evaluate the increase of mitochondrial biogenesis before the mitochondrial transplantation. Finally, the anti-inflammatory effects of these enhanced mitochondria were evaluated after transplantation on LPS-induced in vitro sepsis model.
As authors suggest, data presented are preliminary and the study requires further investigation. However, the manuscript is well designed from an experimental perspective. The experimental hypothesis is clear and the presented data are interesting. The design framework is well defined, but some clarifications should be made to strengthen it.
- To enhance the mitochondrial functionality authors test different molecules at a specific concentration (Results reported in Figure 2). Is it possible that different results could be found by increasing the concentrations of molecules for which an improvement in the cellular OCR was not observed? Did authors try other concentrations of the tested molecules? Authors should clarify this point in the text.
- Authors examined the TNF-α expression levels in LPS-stimulated THP-1 cells by ELISA experiment (Results reported in Figure 4). Authors should indicate whether the reported values have been normalized to the number of cells used for the experiment. Was any normalization protocol on the cellular extracts performed for the obtained values? This should be indicated in the text and in the figure.
- Moreover in the Figure 4 authors should clarify the type of Control they used in the experiment.
- Lines 37-38-101: It is possible that references 3, 4, 1 did not insert in the right position in the text. Authors should check them.
Author Response
Comments and Suggestions for Authors
In the brief report titled “Enhancing mitochondrial function through pharmacological modification: A novel approach to mitochondrial transplantation in a sepsis model” authors describe a strategy to improve the mitochondrial function before transplantation into a cellular model of sepsis. For this strategy, several molecules were tested on rat myoblast L6 cell line and the effect on the oxygen consumption rate (OCR) was evaluated. The best molecule, Formoterol, was used to measure the expression level of mitochondrial-encoded genes in order to evaluate the increase of mitochondrial biogenesis before the mitochondrial transplantation. Finally, the anti-inflammatory effects of these enhanced mitochondria were evaluated after transplantation on LPS-induced in vitro sepsis model.
As authors suggest, data presented are preliminary and the study requires further investigation. However, the manuscript is well designed from an experimental perspective. The experimental hypothesis is clear and the presented data are interesting. The design framework is well defined, but some clarifications should be made to strengthen it.
- To enhance the mitochondrial functionality authors test different molecules at a specific concentration (Results reported in Figure 2). Is it possible that different results could be found by increasing the concentrations of molecules for which an improvement in the cellular OCR was not observed? Did authors try other concentrations of the tested molecules? Authors should clarify this point in the text.
--> We appreciate your valuable comment. We totally agree with you in the point that different results could be found by increasing the concentrations of molecules for which an improvement in the cellular OCR was not observed. So, we added the limitation as follows; Lastly, we chose optimal drug using OCR data, but there might be a chance of mitigating immune dysfunction regardless of OCR data. However, we think that mitochondrial transplantation has impact on inflammation by enhancing mitochondrial function, so we adopted OCR for screening method. We also tested various concentration of candidate drugs, and provided supplementary explanation as follows:
Line 161. when using various concentrations of each drug (data not shown).
2. Authors examined the TNF-α expression levels in LPS-stimulated THP-1 cells by ELISA experiment (Results reported in Figure 4). Authors should indicate whether the reported values have been normalized to the number of cells used for the experiment. Was any normalization protocol on the cellular extracts performed for the obtained values? This should be indicated in the text and in the figure.
--> We appreciate your valuable comment. TNF-alpha levels were calculated using the ELISA standard curve without further normalization. However, equal cell numbers were used in all conditions to ensure comparability.
3. Moreover in the Figure 4 authors should clarify the type of Control they used in the experiment.
--> We appreciate your good comment. We added content for control group in Line 187. The control group consisted of THP-1 cells that were left untreated. We also added this to figure 4 legend.
4. Lines 37-38-101: It is possible that references 3, 4, 1 did not insert in the right position in the text. Authors should check them.
-->We apologize for our mistakes and the incorrect insertion. We changed as follows:
Recent advances in clinical and experimental sepsis research have led to significant discoveries regarding immunometabolism and bioenergetics [3]. Despite the implementation of conventional management approaches such as prompt antibiotic administration and fluid resuscitation, therapeutic interventions remain limited [4].
The position of reference number 3 was changed.
Reference 4 was replaced with a different paper.
Mitochondrial isolation was performed as previously described [25].
Reference number was changed from 1 to 5.
Round 2
Reviewer 1 Report
Comments and Suggestions for Authors
The revised version is very much improved.
Reviewer 2 Report
Comments and Suggestions for Authors
Authors have adequately clarified the not completely clear points that had been identified in the first revision of the manuscript.